# Liganded T3 receptor β2 inhibits the positive feedback autoregulation of the gene for GATA2, a transcription factor critical for thyrotropin production

Naoko Hirahara[1]☯, Hiroko Misawa Nakamura[2]☯, Shigekazu Sasaki[2]*, Akio Matsushita[2], Kenji Ohba[3], Go Kuroda[2], Yuki Sakai[2], Shinsuke Shinkai[2], Hiroshi Haeno[4], Takuhiro Nishio[5], Shuichi Yoshida[5], Yutaka Oki[6], Takafumi Suda[2]

1 Division of Endocrinology and Metabolism, Department of Internal medicine, Japanese Red Cross Shizuoka Hospital, Shizuoka, Shizuoka, Japan, 2 Second Division, Department of Internal Medicine, Hamamatsu University School of Medicine, Hamamatsu, Shizuoka, Japan, 3 Medical Education Center, Hamamatsu University School of Medicine, Hamamatsu, Shizuoka, Japan, 4 Department of Computational Biology and Medical Sciences, Graduate School of Frontier Sciences, The University of Tokyo Kashiwa, Kashiwa, Chiba, Japan, 5 Department of Integrated Human Sciences, Hamamatsu University School of Medicine, Hamamatsu, Shizuoka, Japan, 6 Department of Family and Community Medicine, Hamamatsu University School of Medicine, Hamamatsu, Shizuoka, Japan

☯ These authors contributed equally to this work.
* sasakis@hama-med.ac.jp

**Data Availability Statement:** All relevant data are within the manuscript and its Supporting Information files.

## Abstract

The serum concentration of thyrotropin (thyroid stimulating hormone, TSH) is drastically reduced by small increase in the levels of thyroid hormones (T3 and its prohormone, T4); however, the mechanism underlying this relationship is unknown. TSH consists of the chorionic gonadotropin α (CGA) and the β chain (TSHβ). The expression of both peptides is induced by the transcription factor GATA2, a determinant of the thyrotroph and gonadotroph differentiation in the pituitary. We previously reported that the liganded T3 receptor (TR) inhibits transactivation activity of GATA2 via a tethering mechanism and proposed that this mechanism, but not binding of TR with a negative T3-responsive element, is the basis for the T3-dependent inhibition of the TSHβ and CGA genes. Multiple GATA-responsive elements (GATA-REs) also exist within the GATA2 gene itself and mediate the positive feedback autoregulation of this gene. To elucidate the effect of T3 on this non-linear regulation, we fused the GATA-REs at -3.9 kb or +9.5 kb of the GATA2 gene with the chloramphenicol acetyltransferase reporter gene harbored in its 1S-promoter. These constructs were co-transfected with the expression plasmids for GATA2 and the pituitary specific TR, TRβ2, into kidney-derived CV1 cells. We found that liganded TRβ2 represses the GATA2-induced transactivation of these reporter genes. Multi-dimensional input function theory revealed that liganded TRβ2 functions as a classical transcriptional repressor. Then, we investigated the effect of T3 on the endogenous expression of GATA2 protein and mRNA in the gonadotroph-derived LβT2 cells. In this cell line, T3 reduced GATA2 protein independently of the ubiquitin proteasome system. GATA2 mRNA was drastically suppressed by T3, the concentration of which corresponds to moderate hypothyroidism and euthyroidism. These results

**Funding:** This work was supported in part by a Grant-in Aid for Scientific Research to S. S. from the Ministry of Education, Culture, Sports, Science and Technology of Japan (grant number 24590689). The sponsors or funders play no role in the study design, data collection and analysis, decision to publish, or preparation of the manuscript. https://www.jsps.go.jp/j-grantsinaid/

**Competing interests:** The authors have declared that no competing interests exist.

suggest that liganded TRβ2 inhibits the positive feedback autoregulation of the GATA2 gene; moreover this mechanism plays an important role in the potent reduction of TSH production by T3.

## Introduction

Thyrotropin (thyroid-stimulating hormone, TSH) is a pivotal activators of the production of thyroid hormones (T3 and its prohormone, T4) from the thyroid gland. TSH is a heterodimer consisting of an α chain (chorionic gonadotropin α, CGA) and a β chain (TSHβ) [1]. TSHβ determines the hormonal specificity of TSH whereas CGA heterodimerizes with CGβ, follicle stimulating hormone (FSH) β and luteinizing hormone (LH) β subunits, to form CG, FSH and LH, respectively [2]. Two transcription factors, GATA2 and Pit1, are the critical determinants of thyrotroph differentiation in pituitary [3] and they directly bind with the DNA sequence of the TSHβ promoter [2, 4]. Our previous study revealed that the actual activator of the TSHβ gene is GATA2, whereas Pit1 is necessary for protecting the function of GATA2 from the suppressor region (SR), which is located immediately downstream to the GATA-responsive elements (GATA-REs) [5]. Although detailed analysis has not been performed, hypothyroidism in patients with mutant GATA2 has been reported [6, 7]. Previously we have reported that the signal of thyrotropin releasing hormone (TRH) from hypothalamic paraventricular nucleus (PVN) enhanced the transcriptional activity of GATA2 via the protein kinase C (PKC) pathway [8]. We also found that, although this signaling causes the further stimulation of the TSHβ and CGA expression in the thyrotroph, the T3-dependent repression is dominant over the activation via this pathway.

The mechanisms underlying the T3 receptor (TR)-mediated transcriptional activation of T3-target genes (positive regulation) have been well characterized [9]. In these studies, monkey kidney-derived CV1 cells [10] have often been utilized [11–13]. In contrast, the mechanism underlying the T3-dependent transcriptional repression via TR (negative regulation) has not yet been clarified. TSHβ and CGA genes are the typical genes, which are negatively regulated by T3 [14]. In the mice deficient in TRβ2 [15], which is the pituitary-specific TR, T3-dependent inhibition is ameliorated. Thus, TRβ2 is considered to be the major TR subtype mediating the negative regulation of the TSHβ and CGA genes. As an analogy of the T3-responsive element (TRE) in the positively regulated genes, the presence of the so-called negative TRE (nTRE) in the TSHβ gene has long been postulated [1, 16]. Previously we reported that T3-dependent inhibition of the TSHβ promoter is readily detected even in CV1 cells as long as GATA2, Pit1 and TRs are co-expressed [17]. Unexpectedly, this system revealed that inhibition of the TSHβ gene by T3 is maintained even after the deletion or mutation of the reported nTRE [18]. Thus, instead of the nTRE model, we proposed the tethering model in which TRβ2 interacts with GATA2 Zn-fingers via protein-protein interaction and interferes with the transcriptional function of GATA2 in a T3-dependent manner [13, 18], as observed in the case of inhibition of NFkB-induced transactivation by liganded glucocorticoid receptor [19]. The CGA promoter is also stimulated by GATA2 via its GATA-RE and inhibited by liganded TRβ2 [8, 18]. Thus, in the transcriptional regulation of the TSHβ and CGA genes, GATA2 functions as the platform to select either the activating signal by TRH or inhibitory signal by liganded TRβ2 [8, 13].

Importantly, the serum TSH concentrations are drastically reduced by a small increase of thyroid hormone levels [20]. In the clinical studies, values of serum TSH and thyroid

hormones are usually plotted in a log scale and linear scale, respectively [21–24]. Thus, this feature, often referred to as a "log-linear relationship", makes the serum TSH concentration a very sensitive indicator of thyroid gland function [25]. Although numerous mathematical models [26–28] and the involvement of genetic factors [29] have been proposed, there are few experimental systems that enable us to explore the molecular mechanism underlying this unique relationship between them. GATA2 is expressed not only in thyrotroph but also in gonadotroph [2, 3, 30] and placenta [31–33], resulting in the potentiation of the CGA gene expression [34, 35]. Using trophoblast stem cell line, Rcho-1, and TS cells, Ray et al. [36] surveyed approximately 100-kb regions of the rat and mouse GATA loci using quantitative chromatin immunoprecipitation (ChIP) assay. They demonstrated that the GATA2 gene is transactivated by its own translation product (i.e., GATA2 protein) via multiple GATA-REs in this gene (Fig 1A), as in the case of hematopoietic cell lineage [37–39]. Hence, GATA2 expression is controlled by intracellular positive feedback autoregulation [39]. This mechanism of the GATA2 gene also appears to play a role in pituitary because the thyrotroph and gonadotroph-specific expression of dominant negative-type GATA2 mutant down-regulates the expression of endogenous GATA2 *in vivo* [3]. Given that intracellular positive feedback autoregulation of the GATA2 gene is non-linear, we speculated that inhibition of this control system by T3 may display an inverse non-linear pattern as observed in clinical cases. The GATA2 gene is driven by two promoter, 1S and 1G (Fig 1) [40, 41]. Both promoters are activated by GATA2 via the aforementioned GATA-REs [42] and their transcripts can be detected in the gonadotroph cell line, LβT2 [30, 43, 44]. Here, we constructed the chloramphenicol acetyltransferase (CAT)-reporter genes, in which the 1S-promoter was fused to reported GATA-REs derived from the GATA2 gene [36, 39, 45], and co-transfected them with expression plasmids for GATA2 and TRβ2 into CV1 cells. We found that liganded TRβ2 represses the GATA2-indueced transcriptional function via these GATA-REs. Multi-dimensional input function theory suggested that liganded TRβ2 functions as a classical transcriptional repressor of the GATA2 promoter. Next, we evaluated the effect of T3 on the endogenous GATA2 protein expression in LβT2 cells. We found that T3 reduces GATA2 protein expression independently of the ubiquitin-proteasome system. We observed robust reduction in GATA2 mRNA expression by T3 at concentrations corresponding to those between mild hypothyroidism and euthyroidism. Thus, this study may provide the important insights for the unique relationship between serum TSH and thyroid hormones in vivo.

## Materials and methods

### Plasmid constructions

Expression plasmids for rat TRβ2 (pCMX-rTRβ2) and mouse GATA2 (pcDNA3-mGATA2) have been described previously [8, 17, 18]. Because the firefly luciferase-based reporter gene may be artificially suppressed by liganded TR [[13, 46] and references therein], we employed a CAT-based reporter gene. Three luciferase-based reporter plasmids (generous gifts from Drs. Emery H. Bresnick and Meghan E. Boyer), 1S-Luc (GATA2 gene-derived 1S promoter), (-3.9) 1S-Luc (-3.9 kb GATA-RE fused with 1S-promter) and (+9.5)1S-Luc (+9.5 kb GATA-RE fused with 1S-promter) were amplified by polymerase chain reaction (PCR) using three forward primers, 1Spro-UE2 (5′-gggggaattcgccagaaagcccctgtctggggac-3′), 3.9-UME2 (5′-gggggaattcacgcgaagccgccaggtg-3′) or 9.5-UME2: (5′-ggggg aattcacgcgtccccgcagctaccgggcacccctcctct-3′) and a reverse primer, 1Spro-DNB2 (5′-ggggcatatgagatctgggagacctgagcagtgag-3′). These PCR products were digested by EcoRI and Bgl-II. The human D2 promoter in hD2-CAT [47] was replaced by these PCR products after digestion with restriction enzymes (EcoRI and Bgl-II),

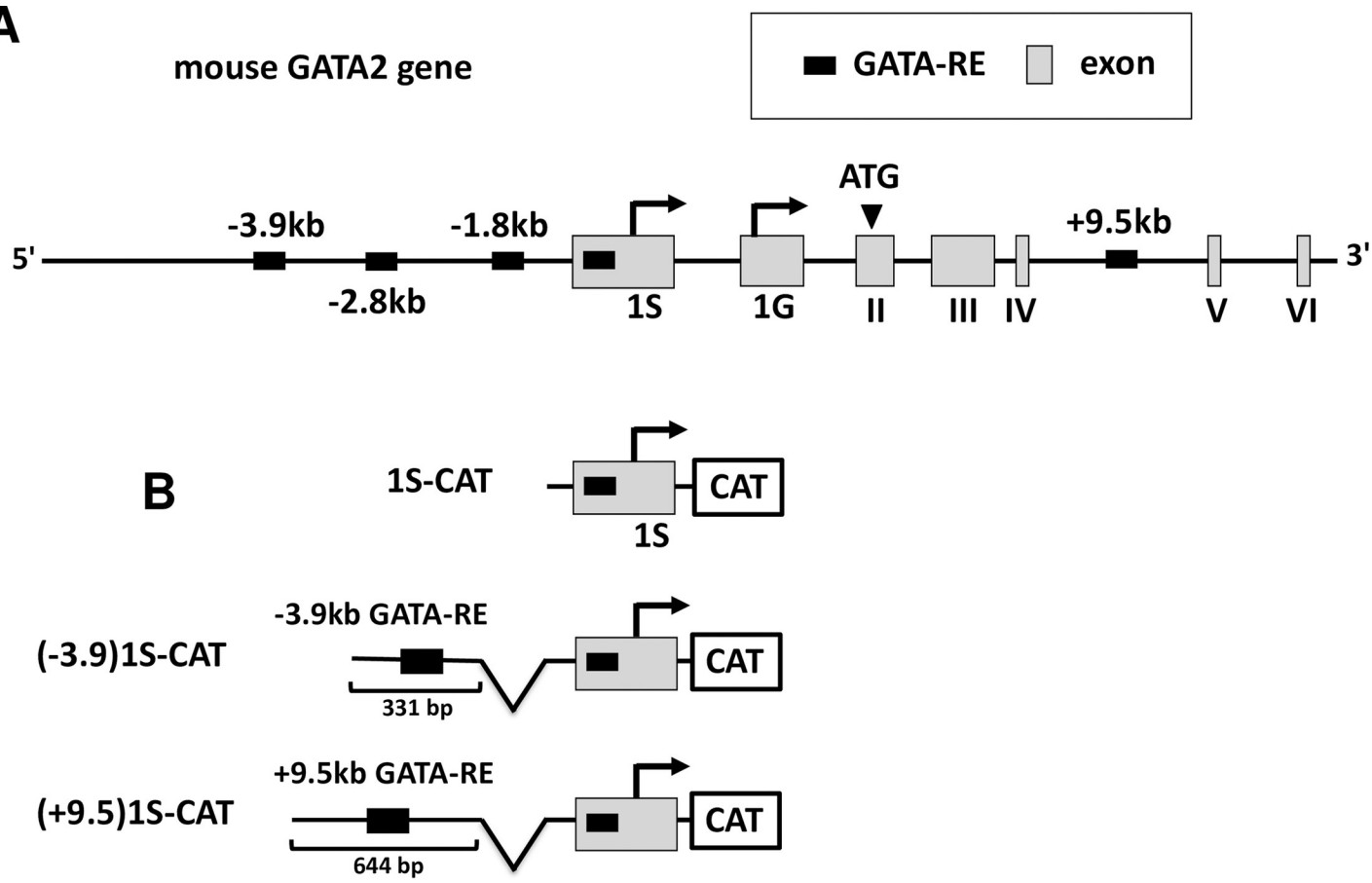

**Fig 1. Structures of the mouse GATA2 gene and CAT-reporter genes.** (A) In the mouse GATA2 gene, there are at least four GATA-REs as well as two promoters (1S and 1G) that are regulated by these GATA-REs. 1S-promoter may also contain a functional GATA-RE (see main text). (B) 1S promoter, promoter fused with -3.9 GATA-RE or promoter fused with +9.5 GATA-RE were subcloned to the CAT reporter gene to generate 1S-CAT, (-3.9)1S-CAT and (+9.5)1S-CAT, respectively.

generating 1S-CAT, (-3.9)1S-CAT and (+9.5)1S-CAT, respectively (Fig 1B). In these constructs, the pUC-derived AP-1-like sequence was deleted because it might also mediate artifactual T3-dependent inhibition [[13, 46] and references therein]. All subcloning sites were confirmed by sequencing.

## Cell culture and transient transfection

CV1 cells [10] were grown in a monolayer culture at 37°C under $CO_2$/air (1:19) in Dulbecco's modified Eagle's medium (DMEM) containing 10% (v/v) fetal calf serum (FCS), penicillin G (100 units/ml), and streptomycin (100 µg/mL). CV1 cells were trypsinized and plated in six-well plates for 24 hr prior to transient transfection using the calcium-phosphate technique [17]. Cells at a density of $2 \times 10^5$ cells per well were transfected with 1.0 µg of the CAT reporter genes (see above), pcDNA3-mGATA2 and pCMX-rTRβ2 together with 0.9 µg of the β-galactosidase expression vector, pCH111 (a modified version of pCH110, Pharmacia LKB Biotechnology, Piscataway, NJ, USA). The total amount of expression plasmid was adjusted with the empty pCMX vector (3.6 µg of DNA in total per dish). After exposure to calcium phosphate/DNA precipitates for 20 hr, the medium was replaced with fresh DMEM containing 10% FCS depleted of thyroid hormones [[17] and references therein] or medium supplemented with T3.

Cells were harvested after incubation for an additional 24 hr, and CAT activity was measured as described previously [17]. Mouse thyrotroph-derived TαT1 cells [48] (a kind gift from Dr. Pamela Mellon, University of California, CA, USA) were seeded on Matrigel-coated plates (Becton Dickinson Labware, Bedford, MA, USA). LβT2 cells, a mouse gonadotroph cell line [43], were cultured in DMEM supplemented with 10% FCS. TαT1 cells and LβT2 cells were maintained under the same conditions as for CV1 cells [17].

### Theoretical characterization of liganded TRβ2 as transcriptional repressor

Based on the theory of multi-dimensional input function (Fig 3, inset) proposed by Alon [49], we calculated the data from CAT reporter assay (Fig 3) in CV1 cells transfected with fixed amount of (+9.5)1S-CAT and pCMX-rTRβ2 together with various amount of pcDNA3-m-GATA2 (0 to 0.2 μg/dish) in the presence of 0–1000 nM T3, using Mathematica software (Champaign, IL, USA). The concentration of liganded TRβ2 was represented as the T3 concentration ([T3]).

### Western blot analysis

Same amounts (200 μl/dish) of whole cell extracts of TαT1 cells or LβT2 cells cultured in a 10 cm dish in the presence or absence of 10 μM MG132 (ChemScene, Monmouth Junction, NJ, USA) and/or T3 for 24 hr were fractionated (20 μl/lane) by sodium dodecyl sulfate polyacrylamide gel electrophoresis (SDS-PAGE), and then, subjected to Western blot analysis with an anti-GATA2 monoclonal antibody (a Kind gift from Drs. Yasuharu Kanki and Tatsuhiko Kodama (University of Tokyo and Perseus Proteomics Inc, Tokyo, Japan)). To assess the levels of transfected GATA2, CV1 cells in a 6-cm dish were transfected with pcDNA3-mGATA2 (5 μg/dish) using calcium-phosphate technique. After incubation for an additional 24 hr, the cells were harvested and subjected to Western blot analysis as mentioned above.

### Reverse transcription-quantitative polymerase chain reaction (RT-qPCR)

LβT2 cells cultured in 10% FCS were incubated with various concentration of T3, and total RNAs was purified using the acid guanidinium thiocyanate-phenol-chloroform extraction method [50]. Total RNA (μg) was mixed with random hexanucleotides and 200 units of Moloney murine leukemia virus reverse transcriptase (Invitrogen Corp., Carlsbad, CA, USA) for first-strand cDNA synthesis. Using the SYBR Green I kit and a LightCycler (Roche Diagnostics, Mannheim, Germany), precipitated cDNA was quantified by real-time PCR using the following primers: forward primer for exon 1G (5'- CACCCCTATCCCGTGAATCCG-3') and reverse primer for exon 2 (5'- AGCTGTGCTGCCTCCATGTAGTTAT-3') [40]. Glyceraldehyde-3-phosphate dehydrogenase (GAPDH) was also amplified from cDNA using a forward primer (5'-TGAACGGGAAGCTCACTGG-3') and reverse primer (5'-TCCACCACCCTGT TGGCTGTA-3'). The thermal cycling conditions were as follows: 10 min at 95˚C, followed by 50 cycles of 10 s at 95˚C for denaturing, 10 s at 62˚C for annealing, and 7 s at 72˚C for extension [50]. PCR signals were quantitatively analyzed using LightCycler software version 3.5 (Roche Diagnostics, Basel, Switzerland).

### Statistical analysis

The CAT reporter assay with CV1 cells and RT-qPCR with LβT2 cells were performed in duplicate three or more times, and each result was expressed as the mean ± S.E. Statistical significance was examined by ANOVA and Fisher's protected least significant difference test

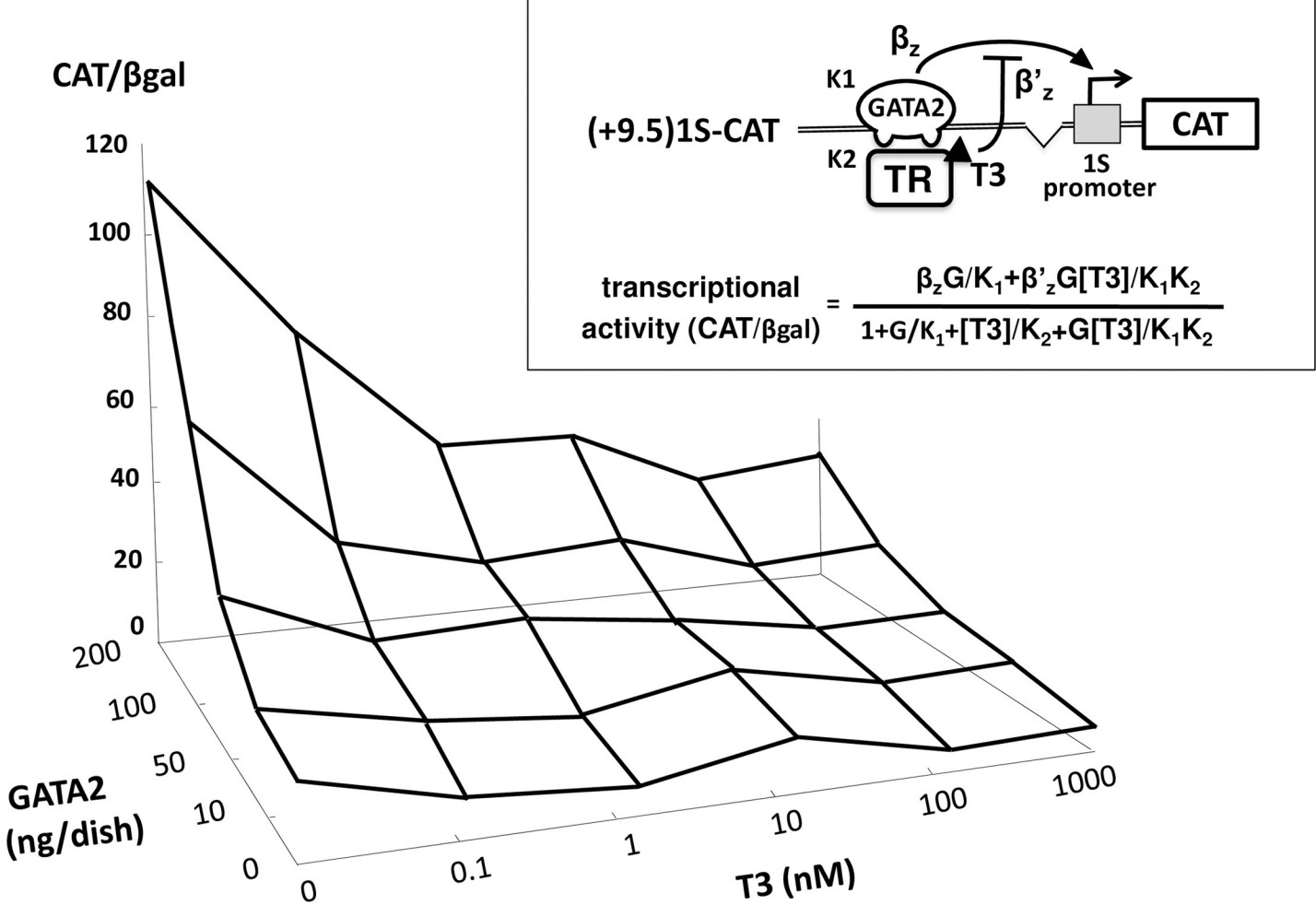

**Fig 3. The liganded TRβ2 functions as a classical transcriptional repressor defined according to multi-dimensional input function theory.** (+9.5)1S-CAT was transfected into CV1 cells along with the expression plasmid for TRβ2 (0.2 μg) and various amounts of that for GATA2 (0 to 200 ng/dish) and treated with various concentrations of T3 (0 to 1000 nM). The data are plotted as a 3-dimesional graph. Results indicate that the liganded TRβ2 tethered to GATA2 Zn-finger functions as a classical transcriptional repressor (inset). The concentration of T3 ([T3]) represents that of liganded TRβ2. Using the data in (A) and the formula for multi-dimensional input function theory (inset) proposed by Alon [49], we calculated the DNA association affinity of GATA2 (K1), liganded TRβ2 (represented as [T3], K2), production rate of the promoter (βz) and basal transcriptional activity (leakage) of the promoter (β'z) as 0.809, 0.081, 544.33 and 129.32, respectively. G, amount of the expression plasmid for GATA2.

using Stat View 4.0 software (Abacus Concepts, Berkeley, CA, USA). A P value <0.05 was considered statistically significant.

## Results

According to the ChIP analysis of the 100-kb regions of the rat and mouse GATA2 loci [36–38], the GATA2 gene harbors multiple GATA-REs (Fig 1A). In trophoblasts [36] and hematopoietic cells [39], GATA2 protein on these DNA elements activates the transcription of this gene from 1S and 1G promoters, resulting in positive feedback autoregulation [39]. Among these GATA-REs, the functions of -3.9 kb and +9.5 kb GATA-REs have been intensively characterized [36, 39, 51]. The expression plasmids of GATA2 and TRβ2 were co-transfected with the CAT reporter genes, of which 1S-promoter was fused to these GATA-REs (Fig 1B) into CV1 cells, which lack endogenous GATA2 [18] or TRs [12]. As shown in Fig 2A, GATA2

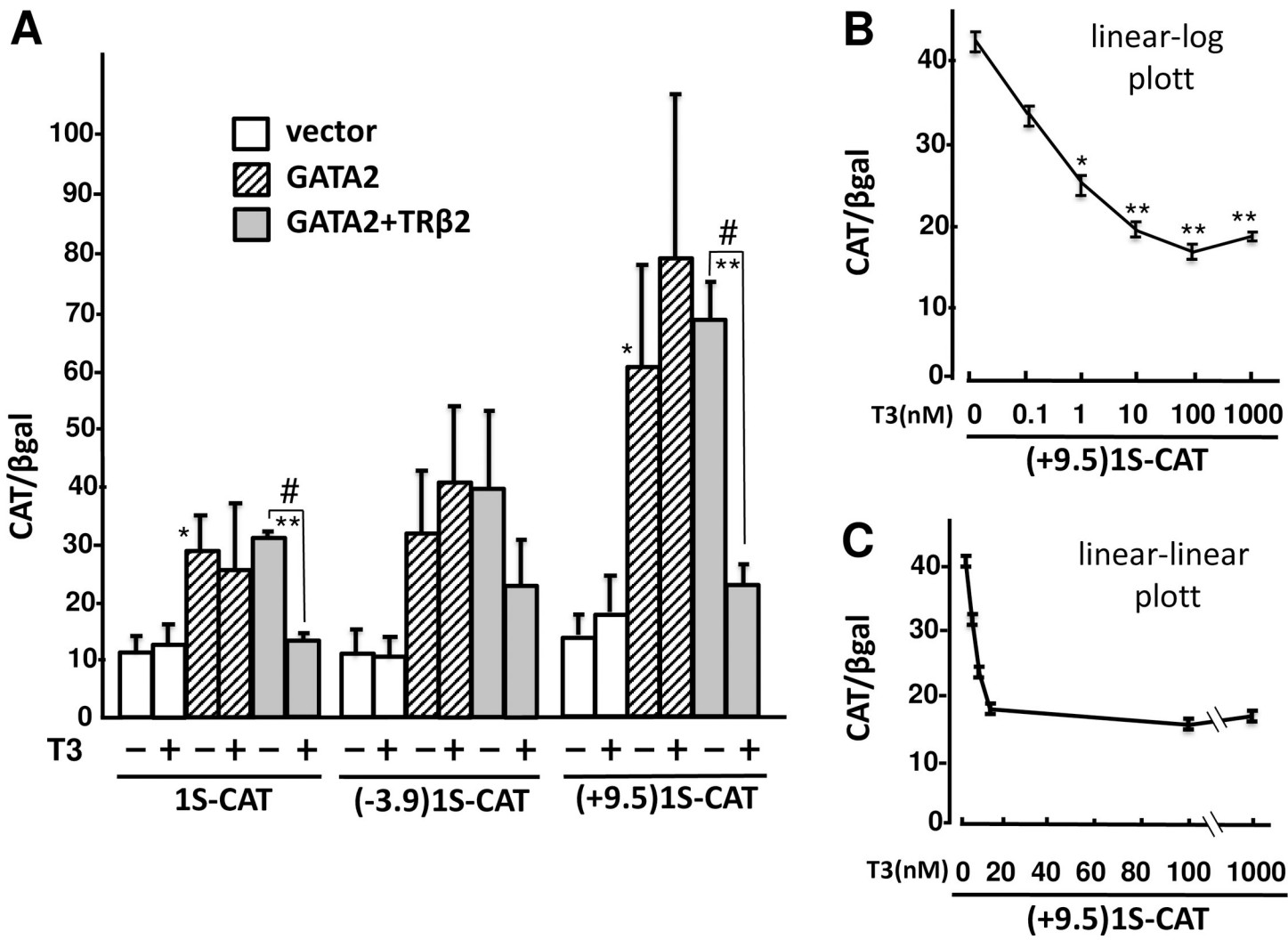

**Fig 2. In CV1 cells, liganded TRβ2 negatively regulates the GATA2-promoter induced by GATA2 itself.** (A) Using the calcium phosphate method, 2.0 μg of 1S-CAT, (-3.9)1S-CAT or (+9.5)1S-CAT was transfected into CV1 cells that were plated at a density of $2\times10^5$ cells per well in a six-well plate along with the expression plasmids for TRβ2 (0.2 μg) and GATA2 (0.1 μg) in the presence or absence of 100 nM T3. *, $P<0.05$ and **, $P<0.01$ for vector vs. GATA2 expression plasmids. #, $P<0.05$ for vehicle vs. 100 nM T3. CAT activity for pCMV-CAT (5.0 ng/well) was taken as 100%. Data are expressed as the mean ± S.E. of three to five independent experiments. (B) (+9.5) 1S-CAT was transfected into CV1 cells as shown in (A) and the cells were treated with various concentration of T3. The results are plotted with [T3] (x-axis) in a log scale and the CAT activity (y-axis) in a linear scale. (C) The data same in (B) are plotted with both [T3] and the CAT activity plotted in a linear scale. *, $P<0.05$ and **, $P<0.01$ vs. T3 (-).

significantly activated 1S-CAT and (+9.5)1S-CAT. The GATA2-dependent activation of 1S-CAT is in agreement with a previous report showing that 1S-promoter also harbors several putative GATA-REs [40]. As shown in Fig 2A, T3 significantly repressed the activities of 1S-CAT and (+9.5)1S-CAT in the presence of GATA2 and TRβ2 (P<0.05). (-3.9)1S-CAT also exhibited a tendency to be activated by GATA2 and suppressed by liganded TRβ2 although the levels of reduction were not statistically significant. As shown in Fig 2B, the transcriptional activity of (+9.5)1S-CAT decreased to approximately 40% by T3 addition. Although Fig 2B was plotted with the T3 concentration ([T3]) in the log scale and CAT activity in the linear scale, drastic repression by T3 was readily observed when both axes were plotted in the linear scale (Fig 2C). Using the experimental system in Fig 2B, we aimed to clarify whether the liganded TR functions as a transcriptional repressor that was defined by the theory of multi-

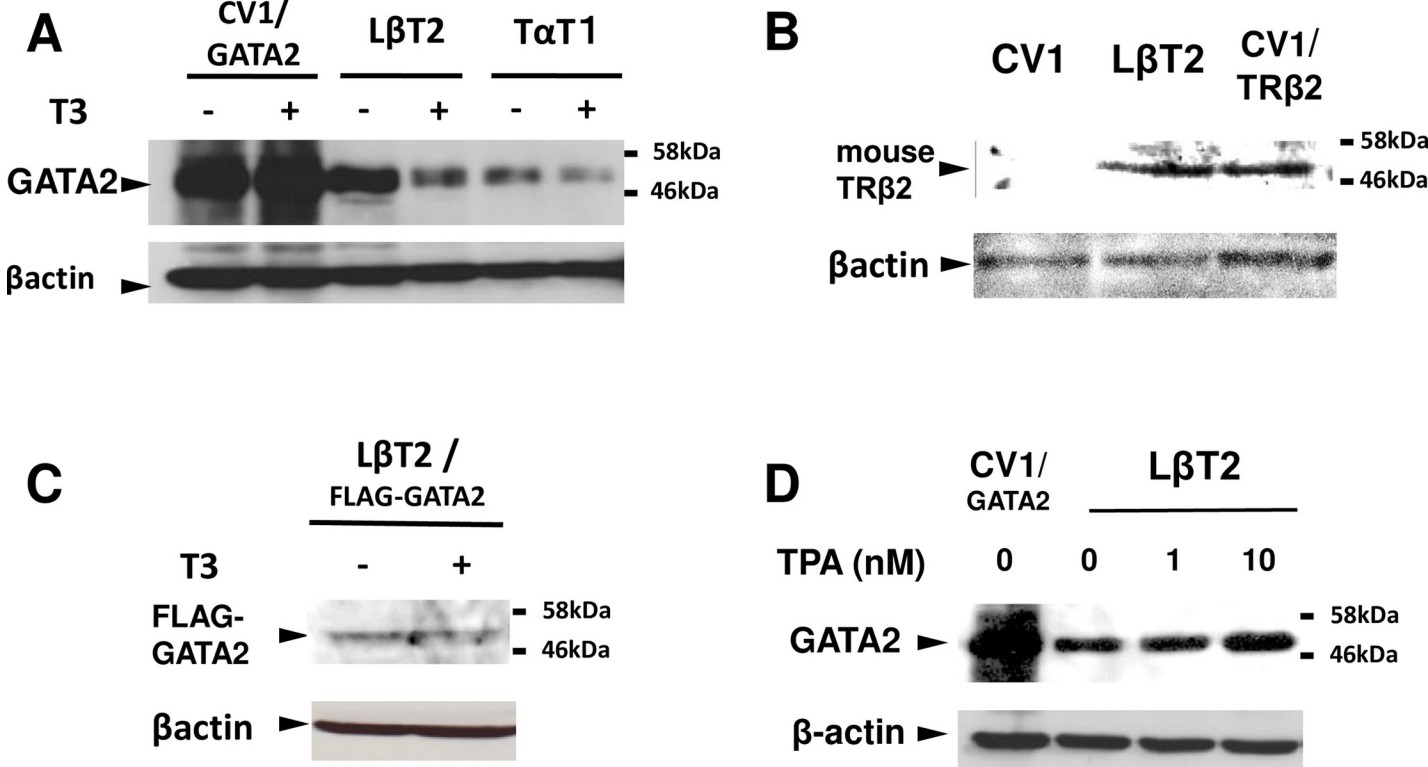

**Fig 4. Endogenous protein expression of GATA2 is inhibited by T3 and enhanced by TPA in LβT2 cells.** (A) Western blot with anti-GATA2 antibody revealed that expression of endogenous GATA2 protein in LβT2 cells is higher than that in TαT1 cells. The GATA2 protein in both cell lines was reduced after 24 hr incubation with 100 nM T3. (B) Western blot with anti-TRβ2 antibody for the whole cell extract of LβT2 cells or that of CV1 cells transfected with expression plasmid of rat TRβ2. (C) FLAG-tagged GATA2 gene driven by the CMV-promoter was transfected into LβT2 cells in 10-cm dish and Western blot with anti-FLAG antibody was performed with the whole cell extract. (D) Western blot with anti-GATA2 antibody for the whole cell extract of LβT2 cells plated in a 10-cm dish treated with 0 to 10 nM TPA for 24 hr.

dimensional input function proposed by Uri Alon (Fig 3 in set) [49]. We co-transfected various amounts of expression plasmids for GATA2 with a fixed amount of (+9.5)1S-CAT into CV1 cells, and evaluated the effect of T3 using 3-dimensional graph (Fig 3). In this setting, we postulated that [T3] represents the concentration of liganded TRβ2. We were able to calculate the promoter association affinity of GATA2 (K1), liganded TRβ2 (K2) represented by [T3], production rate of the promoter (βz) and basal transcriptional activity (leakage) of the promoter (β'z) as 0.809, 0.081, 544.33 and 129.32, respectively. This finding indicated that the liganded TRβ2 tethered to GATA2 behaves as a classical transcriptional repressor.

Next, we examined the inhibitory effect of liganded TRβ2 in the cells, where the GATA2 gene is endogenously expressed presumably via its positive feedback loop. While TαT1 is known to be a thyrotroph cell line [48], its GATA2 protein expression level is modest [8]; we found that the suppressive effect of T3 on TSHβ and the CGA mRNA was mild (data not shown) and completely lost, respectively. Other candidates are gonadotroph-derived LβT2 [43] and choriocarcinoma-derived JEG3 [52], both of which express the CGA gene [30, 34, 35, 52]. Because the expression level of TR in JEG3 cells is very low (personal communication from Dr. Takashi Nagaya, Nagoya University in Japan), we compared expression of the endogenous GATA2 protein in LβT2 cells [53] with that in TαT1 cells by Western blot analysis using anti-GATA2 antibody. As shown in Fig 4A, we observed higher GATA2 protein expression in LβT2 cells than that in TαT1 cells. Both bands were repressed by treatment with 100 nM T3, suggesting the presence of TRs. As expected, we observed endogenous TRβ2 in LβT2 cells (Fig

4B) as observed in TαT1 cells [17]. In contrast, 100 nM T3 did not affect the level of CMV-promoter driven FLAG-tagged GATA2, which was transfected into LβT2 cells and was detected using anti-FLAG antibody (Fig 4C). Based on these findings, we employed LβT2 cells in the subsequent experiments. As is the case of PKC-dependent potentiation of GATA2-induced transactivation in the TSHβ, CGA [8] and vascular adhesion molecule-1 genes [54], 12-O-tetradecanoylphorbol-13-acetate (TPA), a PKC activating compound, enhanced the expression of the GATA2 protein (Fig 4D). Fig 5A and 5B exhibit the T3-dependency and the time course of the GATA2 protein reduction in LβT2 cells. The endogenous GATA2 protein was clearly down regulated after 24 hr treatment with 100 nM T3 (Fig 5B). As GATA2 is known to be rapidly degradated by the ubiquitin-proteasome system [55, 56], we examined the GATA2 protein levels in the presence of MG132, a cell-permeable proteasome inhibitor. While longer exposure of MG132 increased the yield of the GATA2 protein (Fig 5C), T3 decreased it even in the presence of this compound (Fig 5D), suggesting that T3-dependent repression of GATA2 protein occurs prior to degradation by the ubiquitin-proteasome system.

Measurement of the GATA2 mRNA levels enables us to directly evaluate the T3-dependent repression of GATA2 gene transcription without the influence of the ubiquitin-proteasome system. Thus, we performed RT-qPCR to evaluate the endogenous expression of the GATA2 mRNA in LβT2 cells in the presence of various concentration of T3. When GATA2 mRNA standardized by GAPDH mRNA (GATA2/GAPDH ratio) and T3 concentration was plotted in a linear scale and log scale, respectively (Fig 6A), a significant inhibition was observed; the shape of this relationship was a negative sigmoidal curve, suggesting that T3-dependent repression of the GATA2 gene is non-linear. When both values were plotted in a linear scales, robust reduction in the GATA2 mRNA level by T3 was observed (Fig 6B). Moreover, a rigorous decrease in the GATA2/GAPDH ratio by T3 was still observed when the former was plotted in a log scale and the latter was plotted in a linear scale between 0.1–1.0 nM, the concentrations that correspond to those in moderate hypothyroidism and euthyroidism [57],

## Discussion

GATA2 is a critical transcription factor essential for the gene expression of TSHβ and CGA gene [3–5]. As illustrated in Fig 7, we demonstrated here that liganded TRβ2 not only directly inhibits the promoters of the TSHβ and CGA genes [17, 18] but also interferes with the GATA2 promoter. In agreement with our findings, two independent studies reported that GATA2 mRNA expression was increased in the pituitary gland of hypothyroid mice [58, 59]. As shown in Fig 3, the liganded TRβ2 appears to function as a classical transcriptional repressor [49]. However, liganded TRβ2 may cause drastic suppression of the GATA2 gene expression by means of the global interference with the positive feedback loop of this gene (Fig 7), resulting in the maintenance of the homeostasis in hypothalamus-pituitary-thyroid (H-P-T) axis. In mice, deletion of the GATA2 gene specifically in anterior pituitary is not lethal [60] because loss of GATA2 can be partially compensated by the increased expression of the gene for GATA3, another member of the GATA transcription factor family [61]. Interestingly, GATA3 expression may also be controlled via its positive feedback autoregulation [62, 63]. Liganded TRs may also interfere with this GATA3 autoregulation in these mice because they repress GATA3-dependent transactivation [8, 18].

The nTRE was originally defined based on the hypothesis that an unliganded TR may function as transcriptional activator for a gene that is negatively regulated by T3 [16]. However, this hypothesis has been denied because TR-null mice displayed the increased gene expression of the TSHβ or CGA [15, 64–66]. Furthermore, there are several experimental and theoretical issues in the original report of nTREs (reviewed in [13]). Instead, we proposed the tethering

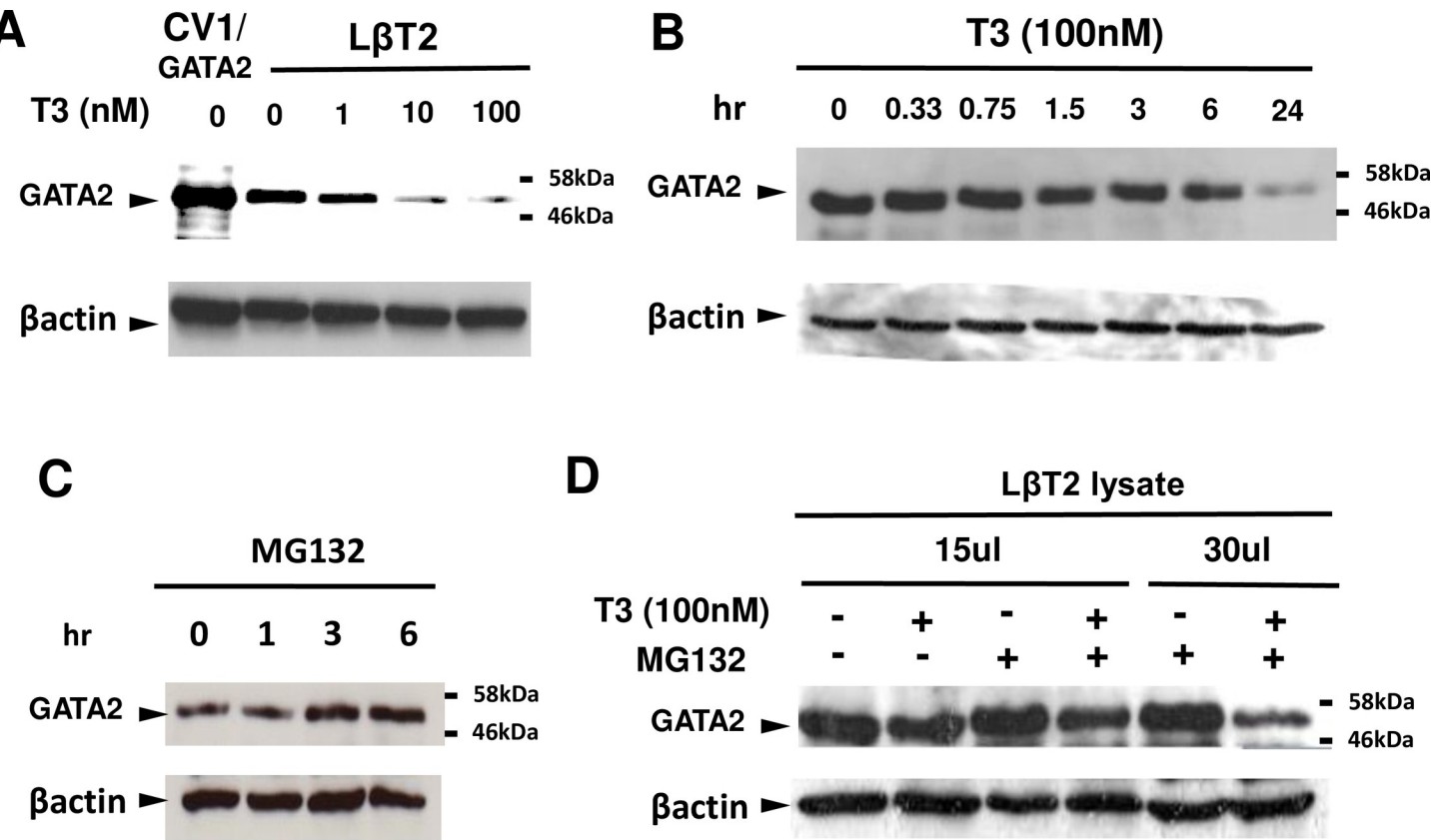

**Fig 5. T3-dependent repression of GATA2 protein occurs independently of the ubiquitin-proteasome system.** (A) Western blot with anti-GATA2 antibody for the whole cell extract of LβT2 cells plated in a 10-cm dish and treated with 0 to 100 nM T3 for 24 hr. CV1/GATA2: CV1 cells transfected with expression plasmid for mouse GATA2. (B) Western blot with anti-GATA2 antibody for the whole cell extract of LβT2 cells plated in a 10-cm dish treated with 0 to 100 nM T3 for 0 to 24 hr. (C) Western blot with anti-GATA2 antibody for the whole cell extract of LβT2 cells plated in a 10-cm dish and treated with 10 μM MG132 for 0 to 6 hr before harvest. (D) Western blot with anti-GATA2 antibody for the whole cell extract (15 μL/lane or 30 μL/lane) of LβT2 cells plated in a 10-cm dish and treated for 24 hr with 10 μM MG132 and 23.5 hr with 100 nM T3 before harvest.

model in which TR associates directly with GATA2 via protein-protein interaction and interferes with GATA2-dependent transactivation in a T3-dependent manner [13, 18]. Although T3 has been known to repress type-2 deiodinase (D2) gene/activity in several organs and cultured cells including thyrotroph [67], no DNA sequence similar to the nTRE reported in the TSHβ gene has been identified in this gene to date [68, 69]. Interestingly, however, the D2 gene harbors the two highly conserved GATA-REs [47]. As expected, our previous work revealed that liganded TRβ2 suppresses the D2 promoter by a tethering mechanism via GATA2 on these GATA-REs [47]. In the GATA2 gene, the critical function of +9.5 kb GATA-RE (Fig 1A) was evidenced based on hematopoietic stem/progenitor cell depletion observed in the mice harboring the deletion of this GATA-RE [45].

Gonadotroph-derived LβT2 cells express the endogenous GATA2 gene (Figs 4 and 5). Thus, these cells may represent a more suitable model than CV1 cells to observe the T3-dependent inhibition of autoregulation of the GATA2 gene. As expected, the graph of T3 inhibition of the GATA2 gene evaluated by RT-qPCR analysis in LβT2 cells is a negative sigmoidal curve (Fig 6A). This pattern of regulation is considered to be suitable for rapid "on and off" control of expression by T3. Fig 6A shows the GATA2/GAPDH ratio (y-axis) in a linear scale and T3 (x-axis) in a log scale, this is the reverse of that seen in clinical situation, i.e., TSH in a log scale and thyroid hormones in a linear scale [21–23, 24]. However, strong repression of GATA2/

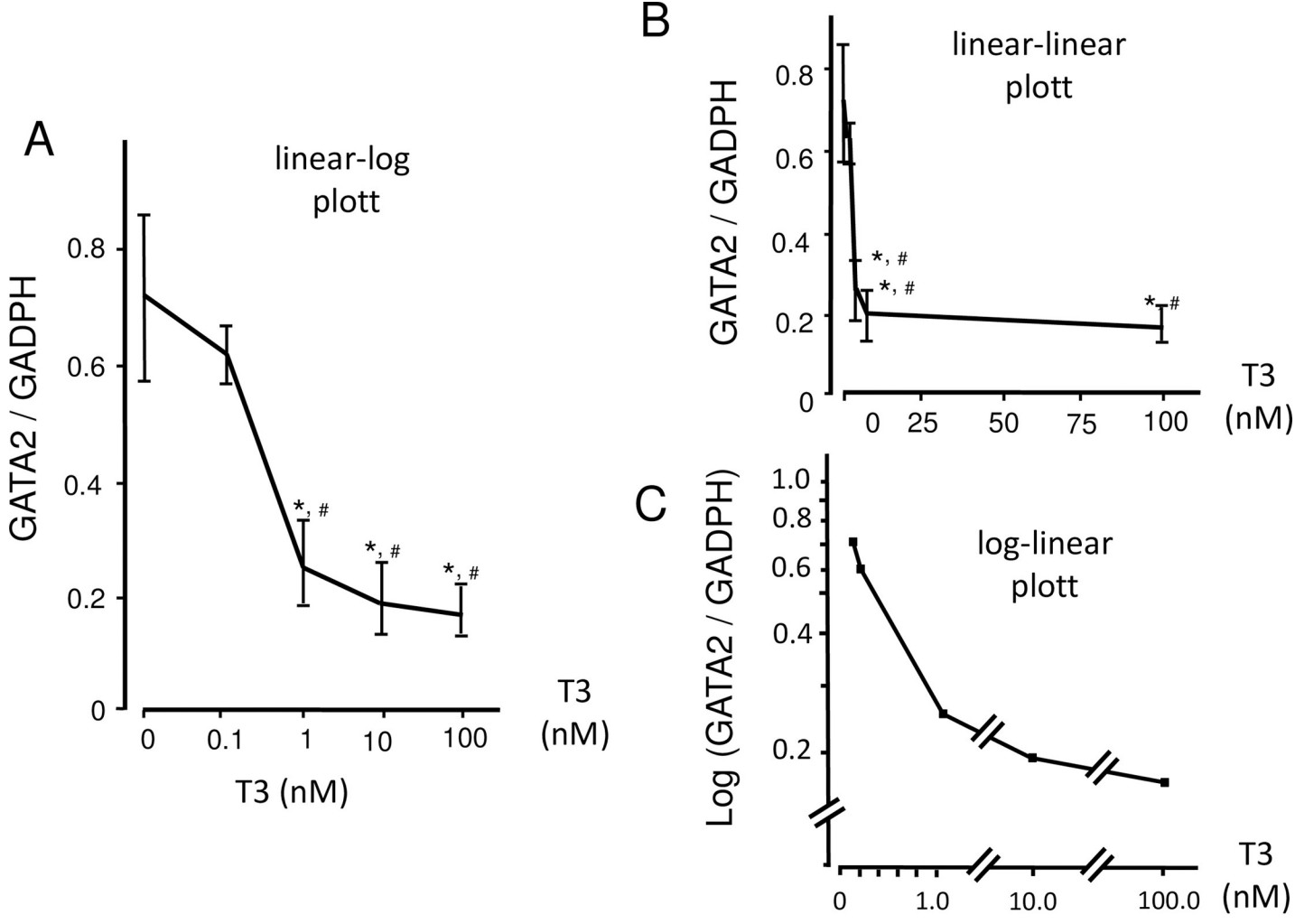

**Fig 6. Inhibition of GATA2 mRNA by T3 in LβT2 cells.** (A) Using RT-qPCR with primers specific for exon 1G and exon 2, the effect of T3 on the expression of GATA2 mRNA in LβT2 cells was evaluated. The shape of this graph was a negative sigmoidal curve. (B) The same data as in (A) are plotted with both [T3] (x-axis) and the GATA2/GAPDH ratio (y-axis) plotted in a linear scale. *, P<0.01 vs. T3(-). #, P<0.01 vs. T3 = 0.1 nM. (C) The same data as in (A) and (B) with the GATA2/GAPDH ratio (y-axis) in a log scale and [T3] (x-axis) between 0–1.0 nM in a linear scale, which corresponds to moderate hypothyroidism and euthyroidism [57].

GAPDH ratio by T3 was observed when both are plotted in a linear scale (Fig 6B). Furthermore, potent suppression was found even when the GATA2/GAPDH ratio and T3 were plotted in a log scale and linear scale, respectively (Fig 6C), especially at the T3 concentration those observed in moderate hypothyroidism and euthyroidism [57]. Thus, in the LβT2 cells, T3-dependent inhibition of positive feedback autoregulation in the GATA2 gene may contribute, at least in part, to the non-linear relationship between TSH production and thyroid hormones in vivo.

In this study, use of cell culture system enabled us to exclude the influence of TRH from hypothalamic PVN. However, the T3-dependent negative regulation of TRH production should be considered [70, 71] because, in thyrotroph, TRH receptor (TRH-R) signaling potentiates the transactivation function of GATA2 via the PKC pathway, resulting in the enhanced expression of the TSHβ and CGA genes [8] as well as the GATA2 gene (Fig 4D). Notably, the alteration in the level of TRH mRNA level in PVN after thyroidectomy and subsequent T3 supplementation are very similar to those of TSHβ mRNA in the pituitary [72]. Moreover, the

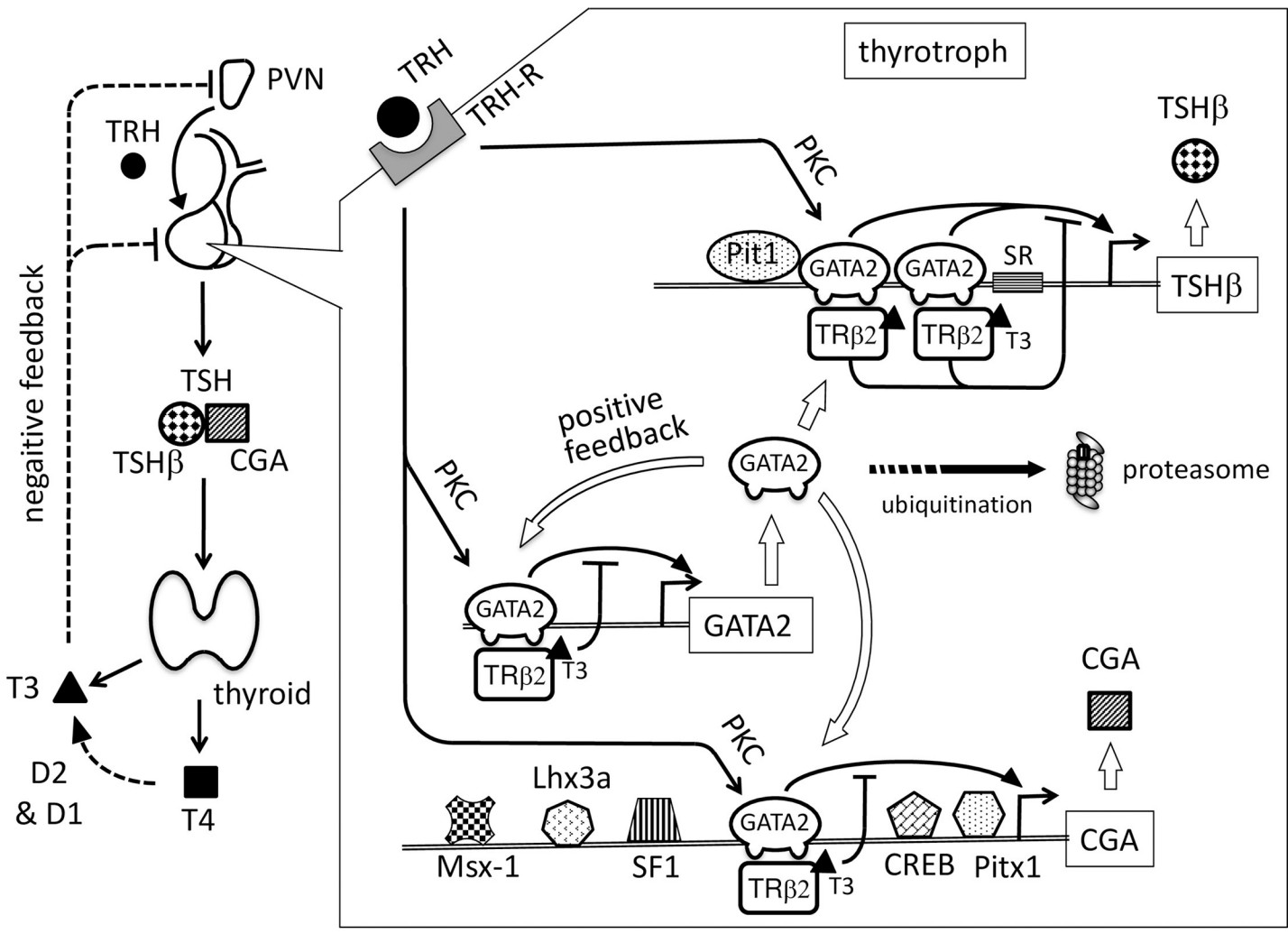

**Fig 7. A schematic representation of hypothalamus-pituitary-thyroid (H-P-T) axis and transcriptional regulation of the GATA2, TSHβ and CGA genes by GATA2 and liganded TRβ2 in thyrotroph.** To simplify, only one GATA-RE is illustrated in the 5' region of the GATA2 gene. In a TRH-dependent manner, TRH-R stimulates the transcriptional activity of GATA2 via the PKC pathway. In contrast, TRβ2 associates with the Zn-finger region of GATA2 via protein-protein interaction, resulting in the T3-dependent interference of GATA2-dependent transactivation (tethering mechanism). Our study suggests that liganded TRβ2 inhibits not only the transcription of the TSHβ and CGA genes but also the positive feedback autoregulation of the GATA2 gene. Thus, there are the intracellular cascades of T3-dependent negative regulation between GATA2 and TSHβ/CGA. Reduction of GATA2 protein expression by T3 appears to occur at transcriptional level and is independent from the ubiquitin-proteasome system. T3 also causes TSHβ mRNA to be unstable. D2, type 2 deiodinase (the major deiodinase in thyrotroph). SR, suppressor region (see main text). In addition to GATA2, the CGA promoter is regulated by multiple transcription factors including Msx1, SF-1, Lhx3a, cAMP response element binding protein (CREB) and Ptx1 [35].

GATA2 and TRβ2 genes are induced in the cultured neurons that are over-expressed with Sim1, a transcription factor that determines the differentiation of the PVN in the hypothalamus [73, 74]. We recently observed that GATA2 is expressed in the TRH neuron of the rat PVN, and found that liganded TRβ2 represses the preproTRH gene activated by GATA2 on its GATA-RE (Kuroda et al. paper in preparation). Thus, T3-dependent inhibition of the positive feedback regulation of the GATA2 gene in both thyrotrophs and the PVN may share common mechanism. Indeed, Fekete et al. [71] previously reported an inverse log-linear relationship between the numbers of preproTRH positive cells in rat PVN as evaluated by in situ hybridization analysis and plasma T3 concentration. Interestingly, detailed analysis of 152261 human subjects revealed that the relationship between TSH in a log scale and free T4 in a linear scale

may not be inverse log-linear but can be described by two overlapping negative sigmoidal curves [24].

Several additional mechanisms are considered to be involved in the negative feedback regulation of the H-P-T axis. First, although we focused on the feedback regulation of the GATA2 gene, T3-dependent inhibition may be amplified by the intracellular cascades between the GATA2 gene and the TSHβ and CGA genes (Fig 7). Second, T3 also represses the expression of prohormone convertase 1/3 and 2, which play essential roles in the processing of the proTRH peptide in the PVN [75], and the TRH-R expression in pituitary [76, 77]. Third, T3 destabilizes the TSHβ mRNA but not CGA mRNA [78]. Fourth, T3 may inhibit the expression of Pit1 [79], which is a critical transcription factor for the TSHβ gene (Fig 7), but not the CGA gene [2]. Finally, the cell cycle of thyrotrophs [80, 81] may also be involved because sustained hypothyroidism often causes thyrotroph hyperplasia, resulting in the massive enlargement of the anterior pituitary [82].

GATA2 was originally identified as an essential transcription factor that mediates differentiation of hematopoietic cells [61]. Consistently, hematopoietic cell differentiation in mice is affected by T3 administration [83]; moreover, it is impaired in the mice that harbor genetic defects for the gene coding TRα [83, 84], the major TR subtype in the erythroid cell lineage. In human subjects with mutations in the TRα gene, anemia and erythroid defect were reported [85]. Thus, the precise mechanism of the effect of T3 on the GATA2-related hematopoiesis should be investigated in future. Current results also predict that serum TSH level may be correlated with placental function because this organ expresses GATA family members [32, 33] as well as TRs [86]. As is the case with the TSHβ and CGA gene in the pituitary (Fig 7), GATA2 and GATA3 regulate the expression of the genes that encode the critical proteins responsible for maintaining the placental function, including the component of bone morphogenetic protein 4, Nodal and Wnt signals [32]. These findings may provide the insight into why subtle elevation in the serum TSH concentration ($> 2.5$ μIU/L) is correlated with the pregnancy loss rate in thyroid antibody-negative women [87–89].

## Supporting information

**S1 Raw Images.**
(PDF)

## Acknowledgments

We are grateful to the following investigators for providing plasmids: Drs. Meghan E. Boyer and Emery H. Bresnick (University of Wisconsin, USA), Kazuhiko Umesono (Kyoto University, Japan) and Masayuki Yamamoto (Tsukuba University, Japan). We also thank Drs. Yasuharu Kanki and Tatsuhiko Kodama (University of Tokyo) for providing the anti-GATA2 antibody and Dr. Pamela Mellon (University of California, CA, USA) for the TαT1 and LβT2 cells.

## Author Contributions

**Conceptualization:** Shigekazu Sasaki.

**Data curation:** Hiroko Misawa Nakamura, Shigekazu Sasaki.

**Formal analysis:** Hiroko Misawa Nakamura, Shigekazu Sasaki, Akio Matsushita, Kenji Ohba, Go Kuroda, Hiroshi Haeno, Takuhiro Nishio, Shuichi Yoshida, Yutaka Oki, Takafumi Suda.

**Funding acquisition:** Shigekazu Sasaki.

**Investigation:** Naoko Hirahara, Hiroko Misawa Nakamura, Shigekazu Sasaki, Go Kuroda, Yuki Sakai, Shinsuke Shinkai.

**Methodology:** Naoko Hirahara, Hiroko Misawa Nakamura, Akio Matsushita, Hiroshi Haeno.

**Project administration:** Takafumi Suda.

**Resources:** Shigekazu Sasaki, Akio Matsushita, Go Kuroda, Yuki Sakai, Shinsuke Shinkai.

**Software:** Takuhiro Nishio, Shuichi Yoshida.

**Supervision:** Shigekazu Sasaki.

**Validation:** Shigekazu Sasaki, Akio Matsushita, Kenji Ohba, Go Kuroda, Yuki Sakai, Shinsuke Shinkai, Hiroshi Haeno, Takuhiro Nishio, Shuichi Yoshida, Yutaka Oki, Takafumi Suda.

**Visualization:** Shigekazu Sasaki.

**Writing – original draft:** Naoko Hirahara, Hiroko Misawa Nakamura, Shigekazu Sasaki.

**Writing – review & editing:** Naoko Hirahara.

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
