## [Decision Letter · Decision Letter 0]

3 Oct 2019

PONE-D-19-22945

Liganded T3 receptor β2 inhibits the positive feedback autoregulation of the gene for GATA2, a transcription factor critical for the thyrotropin production

PLOS ONE

Dear Sasaki,

Thank you for submitting your manuscript to PLOS ONE. After careful consideration, we feel that it has merit but does not fully meet PLOS ONE’s publication criteria as it currently stands. Therefore, we invite you to submit a revised version of the manuscript that addresses the points raised during the review process.

We would appreciate receiving your revised manuscript by Nov 17 2019 11:59PM. To enhance the reproducibility of your results, we recommend that if applicable you deposit your laboratory protocols in protocols.io, where a protocol can be assigned its own identifier (DOI) such that it can be cited independently in the future. For instructions see: http://journals.plos.org/plosone/s/submission-guidelines#loc-laboratory-protocols

We look forward to receiving your revised manuscript.

Kind regards,

Hiroyoshi Ariga

Academic Editor

PLOS ONE

**Journal requierments**

**Comments to the Author**

1. Is the manuscript technically sound, and do the data support the conclusions?

Reviewer #1: Yes

2. Has the statistical analysis been performed appropriately and rigorously? 

Reviewer #1: Yes

3. Have the authors made all data underlying the findings in their manuscript fully available?

Reviewer #1: Yes

4. Is the manuscript presented in an intelligible fashion and written in standard English?

Reviewer #1: Yes

5. Review Comments to the Author

Reviewer #1: Hirahara et al studied GATA2 promoter function as well as expression of endogenous GATA2 in cultured cells. They demonstrated that GATA2 promoter is down regulated by T3 in the presence of T3Rß2. They also showed endogenous GATA2 is down regulated in LßT2 cells in a manner which is not inhibited by MG132. In general the experiments are well designed and quality of the data is good. This reviewer have one major comment and several minor.

Major. Fig.2 T3 is added only in GATA2+ T3Rß2 cells. Effect of T3 should be analyzed (although the authors may have shown similar data in previous papers), in cells with vector only and GATA2 only.

Minor.

p. 3 L83, “has not performed” should be “has not been performed”

p.6 L190, p.7 L198, 201, period is not required for “hr” or they should be replaced to “hrs”.

L 222, same comment.

L225-227, one closing bracket is lacking.

p.9 284, “theoretically”?, shouldn’t it be “experimentally”? Please rephrase the sentence.

Also this reviewer would like to confirm the authors whether “no study showed that liganded TR functioned as a transcriptional repressor”, including the authors’ preceding studies.

Or, these statements should be moved to after the sentence in L290-295, to make sense.

p.11 L344 “ as log scale” should be “ in a log scale”.

As it appears that articles are lacking or inappropriate here and there throughout the manuscript, further English editing is recommended.

p.14, l430 to later,

“Several mechanisms” should be explained more clearly. The sentences following “ first, seconde,…” should simply state the mechanism, but not the reasoning.

For an example, “T3 may destabilize TSH beta mRNA” should follow immediately after “Third”…something like that. Employing LßT2 cells cannot be one of the additional mechanisms.

6. PLOS authors have the option to publish the peer review history of their article (what does this mean?). If published, this will include your full peer review and any attached files.

Reviewer #1: No

---

## [Author Response · Author response to Decision Letter 0]

23 Nov 2019

REFERENCE: PONE-D-19-22945

Title: "Liganded T3 receptor β2 inhibits the positive feedback autoregulation of the gene for GATA2, a transcription factor critical for thyrotropin production”

AUTHORS: Naoko Hirahara and Hiroko Misawa Nakamura et al. 

PLOS ONE

Academic Editor: Hiroyoshi Ariga

Please re-evaluate our revised manuscript. Although our previous manuscript was screened once by a native English speaker before first submission, the referee recommended us to edit it again regarding in particular to the usage of articles. So we asked another (more experienced) person to re-check it. Based on his recommendation, we deleted “the” from “the thyrotropin production” in the title. If it can be now acceptable in your journal, we would be very much pleased. Thank you for your kind arrangement. 

Shigekazu Sasaki. MD.

Senior Assistant Professor

Second Division of Internal Medicine

Hamamatsu University School of Medicine

1-20-1 Handayama Higashi-ku 

Hamamatsu 431-3192, Japan

E-mail: sasakis@hama-med.ac.jp

RESPONSE TO THE COMMENTS FROM REVIEWER

Major.

Regarding to the effect of T3 on the CAT reporter genes co-transfected with vector only and GATA2 only (Fig.2),

To evaluate the effect of T3 (100 nM) on the activities of CAT reporter genes co-transfected with vector only and GATA2 only, we conducted additional CAT assays for five times independently. Using the value of the activity of the CMV-based CAT reporter gene (inter assay control) as 100, we combined these data with the original data of Fig. 2A and created new Fig. 2A. We did not find any statistical significance in CAT activities between absence or presence of T3 when we co-transfected vector only or GATA2 only. 

Minor.

Regarding to “has not performed” of p. 3 L83 in the previous manuscript,

we added “been”. 

Regarding to period for “hr” in p.6 L190, p.7 L198, p.7 L201 and p.7 L222 of the previous manuscript,

we deleted all the periods for “hr”.

Regarding to lack of a closing bracket in L225-227 of the previous manuscript,

we added a closing bracket. 

Regarding to “theoretically” in p.9 284 of the previous manuscript, 

we rephrased the sentence as follows.

Using the experimental system in Fig. 2B, we wanted to know whether the liganded TR functions as a transcriptional repressor that was defined by the theory of multi-dimensional in put function proposed by Uri Alon (Fig. 3 in set) [49]. We co-transfected various amounts of---. 

Regarding to “ as log scale” in the previous manuscript,

we corrected “ as log scale” to “ in a log scale”

Regarding to lacking or inappropriate usage of articles,

Thank you for your recommendation. Although a native English speaker screened our previous manuscript once before submission, we asked another (more experienced) person to re-check English including usage of articles. He pointed a lot of mistakes in the previous manuscript. We corrected all of them. In addition, based on his recommendation, we deleted “the” from “the thyrotropin production” in the title (Please see “Track Changes”). We appreciate your recommendation very much. 

Regarding to “Several mechanisms” in p.14, l430 of the previous manuscript,

We altered the sentences as follows.

Third, T3 destabilizes the TSHβ mRNA but not CGA mRNA [78]. Fourth, T3 may inhibit the expression of Pit1 [79], which is a critical transcription factor for the TSHβ gene (Fig.7), but not the CGA gene [2]. Finally, the cell cycle of thyrotrophs [80, 81] may also be involved because sustained hypothyroidism often causes thyrotroph hyperplasia, resulting in the massive enlargement of the anterior pituitary [82].

In the course of this revision, we found two mistakes in p.18 L554 and L556: the phrase “the CAT activity” in these sentences was corrected to “the GATA2/GAPDH ratio”.

---

## [Decision Letter · Decision Letter 1]

29 Nov 2019

PONE-D-19-22945R1

Liganded T3 receptor β2 inhibits the positive feedback autoregulation of the gene for GATA2, a transcription factor critical for thyrotropin production

PLOS ONE

Dear Dr. Sasaki,

Thank you for submitting your manuscript to PLOS ONE. After careful consideration, we feel that it has merit but does not fully meet PLOS ONE’s publication criteria as it currently stands. Therefore, we invite you to submit a revised version of the manuscript that addresses the points raised during the review process.

We would appreciate receiving your revised manuscript by Jan 13 2020 11:59PM. To enhance the reproducibility of your results, we recommend that if applicable you deposit your laboratory protocols in protocols.io, where a protocol can be assigned its own identifier (DOI) such that it can be cited independently in the future. For instructions see: http://journals.plos.org/plosone/s/submission-guidelines#loc-laboratory-protocols

We look forward to receiving your revised manuscript.

Kind regards,

Hiroyoshi Ariga

Academic Editor

PLOS ONE

Reviewers' comments:

Reviewer's Responses to Questions

**Comments to the Author**

1. If the authors have adequately addressed your comments raised in a previous round of review and you feel that this manuscript is now acceptable for publication, you may indicate that here to bypass the “Comments to the Author” section, enter your conflict of interest statement in the “Confidential to Editor” section, and submit your "Accept" recommendation.

Reviewer #1: (No Response)

2. Is the manuscript technically sound, and do the data support the conclusions?

Reviewer #1: Yes

3. Has the statistical analysis been performed appropriately and rigorously? 

Reviewer #1: Yes

4. Have the authors made all data underlying the findings in their manuscript fully available?

Reviewer #1: Yes

5. Is the manuscript presented in an intelligible fashion and written in standard English?

Reviewer #1: No

6. Review Comments to the Author

Reviewer #1: The authors performed additional experiments according to the query raised by this reviewer. In addition, the manuscript has been improved considerably by further editing. This manuscript is almost acceptable, however, I still found some points that should be corrected.

Line 43, “a determinants” should be “ a determinant” or “one of the determinants”

Line 288, “ wanted to know” in an original scientific article sounds a bit unusual to me. Maybe, “aimed to clarify” or “tried to evaluate” should be suitable.

7. PLOS authors have the option to publish the peer review history of their article (what does this mean?). If published, this will include your full peer review and any attached files.

Reviewer #1: No

---

## [Author Response · Author response to Decision Letter 1]

20 Dec 2019

REFERENCE: PONE-D-19-22945

Title: "Liganded T3 receptor _2 inhibits the positive feedback autoregulation of the gene for GATA2, a transcription factor critical for thyrotropin production”

AUTHORS: Naoko Hirahara and Hiroko Misawa Nakamura et al. 

PLOS ONE

Academic Editor: Hiroyoshi Ariga

Please re-evaluate our revised manuscript. We corrected the sentences that the referee pointed. In addition, we transferred the original figure file to seven separate tif-style files using PACE home page (https://pacev2.apexcovantage.com/). If it can be now acceptable in your journal, we would be very much pleased. Thank you for your kind arrangement. 

Shigekazu Sasaki. MD.

Senior Assistant Professor

Second Division of Internal Medicine

Hamamatsu University School of Medicine

1-20-1 Handayama Higashi-ku 

Hamamatsu 431-3192, Japan

E-mail: sasakis@hama-med.ac.jp

RESPONSE TO THE COMMENTS FROM REVIEWER

We corrected the sentences that the referee pointed as follows. 

Line 43, “a determinants” 

We corrected as “ a determinant” 

Line 288, “ wanted to know” 

We corrected as “aimed to clarify”.

---

## [Decision Letter · Decision Letter 2]

26 Dec 2019

Liganded T3 receptor β2 inhibits the positive feedback autoregulation of the gene for GATA2, a transcription factor critical for thyrotropin production

PONE-D-19-22945R2

Dear Dr. Sasaki,

We are pleased to inform you that your manuscript has been judged scientifically suitable for publication and will be formally accepted for publication once it complies with all outstanding technical requirements.

With kind regards,

Hiroyoshi Ariga

Academic Editor

PLOS ONE

Reviewers' comments:

Reviewer's Responses to Questions

**Comments to the Author**

1. If the authors have adequately addressed your comments raised in a previous round of review and you feel that this manuscript is now acceptable for publication, you may indicate that here to bypass the “Comments to the Author” section, enter your conflict of interest statement in the “Confidential to Editor” section, and submit your "Accept" recommendation.

Reviewer #1: All comments have been addressed

2. Is the manuscript technically sound, and do the data support the conclusions?

Reviewer #1: Yes

3. Has the statistical analysis been performed appropriately and rigorously? 

Reviewer #1: Yes

4. Have the authors made all data underlying the findings in their manuscript fully available?

Reviewer #1: Yes

5. Is the manuscript presented in an intelligible fashion and written in standard English?

Reviewer #1: Yes

6. Review Comments to the Author

Reviewer #1: Hirahara et al studied GATA2 promoter function as well as expression of endogenous GATA2 in cultured cells. They demonstrated that GATA2 promoter is down regulated by T3 in the presence of T3Rß2. They also showed endogenous GATA2 is down regulated in LßT2 cells in a manner which is not inhibited by MG132. In general the experiments are well designed and quality of the data is good.

As the authors adequately addressed to the comments raised by this reviewer, this reviewer feels that the manuscript is now acceptable.

7. PLOS authors have the option to publish the peer review history of their article (what does this mean?). If published, this will include your full peer review and any attached files.

Reviewer #1: No

---

## [Editor Report · Acceptance letter]

30 Dec 2019

PONE-D-19-22945R2 

Liganded T3 receptor β2 inhibits the positive feedback autoregulation of the gene for GATA2, a transcription factor critical for thyrotropin production 

Dear Dr. Sasaki:

I am pleased to inform you that your manuscript has been deemed suitable for publication in PLOS ONE. Congratulations! Your manuscript is now with our production department. 

With kind regards,

on behalf of

Dr. Hiroyoshi Ariga 

Academic Editor

PLOS ONE